# Intracellular Angiotensin II Stimulation of Sodium Transporter Expression in Proximal Tubule Cells via AT_1_ (AT_1a_) Receptor-Mediated, MAP Kinases ERK1/2- and NF-кB-Dependent Signaling Pathways

**DOI:** 10.3390/cells12111492

**Published:** 2023-05-28

**Authors:** Xiaochun Li, Jialong Zhuo

**Affiliations:** 1Tulane Hypertension and Renal Center of Excellence, Tulane University School of Medicine, New Orleans, LA 70112-2699, USA; xli68@tulane.edu; 2Department of Physiology, Tulane University School of Medicine, New Orleans, LA 70112-2699, USA

**Keywords:** intracellular angiotensin II, blood pressure, Na^+^/HCO_3_^-^ transport, proximal tubule

## Abstract

The current prevailing paradigm in the renin-angiotensin system dictates that most, if not all, biological, physiological, and pathological responses to its most potent peptide, angiotensin II (Ang II), are mediated by extracellular Ang II activating its cell surface receptors. Whether intracellular (or intracrine) Ang II and its receptors are involved remains incompletely understood. The present study tested the hypothesis that extracellular Ang II is taken up by the proximal tubules of the kidney by an AT_1_ (AT_1a_) receptor-dependent mechanism and that overexpression of an intracellular Ang II fusion protein (ECFP/Ang II) in mouse proximal tubule cells (mPTC) stimulates the expression of Na^+^/H^+^ exchanger 3 (NHE3), Na^+^/HCO_3_^-^ cotransporter, and sodium and glucose cotransporter 2 (Sglt2) by AT_1a_/MAPK/ERK1/2/NF-kB signaling pathways. mPCT cells derived from male wild-type and type 1a Ang II receptor-deficient mice (*Agtr1a*^-/-^) were transfected with an intracellular enhanced cyan fluorescent protein-tagged Ang II fusion protein, ECFP/Ang II, and treated without or with AT_1_ receptor blocker losartan, AT_2_ receptor blocker PD123319, MEK1/MEK2 inhibitor U0126, NF-кB inhibitor RO 106-9920, or p38 MAP kinase inhibitor SB202196, respectively. In wild-type mPCT cells, the expression of ECFP/Ang II significantly increased NHE3, Na^+^/HCO_3_^-^, and Sglt2 expression (*p* < 0.01). These responses were accompanied by >3-fold increases in the expression of phospho-ERK1/2 and the p65 subunit of NF-кB (*p* < 0.01). Losartan, U0126, or RO 106-9920 all significantly attenuated ECFP/Ang II-induced NHE3 and Na^+^/HCO_3_^-^ expression (*p* < 0.01). Deletion of AT_1_ (AT_1a_) receptors in mPCT cells attenuated ECFP/Ang II-induced NHE3 and Na^+^/HCO_3_^-^ expression (*p* < 0.01). Interestingly, the AT_2_ receptor blocker PD123319 also attenuated ECFP/Ang II-induced NHE3 and Na^+^/HCO_3_^-^ expression (*p* < 0.01). These results suggest that, similar to extracellular Ang II, intracellular Ang II may also play an important role in Ang II receptor-mediated proximal tubule NHE3, Na^+^/HCO_3_^-^, and Sglt2 expression by activation of AT_1a_/MAPK/ERK1/2/NF-kB signaling pathways.

## 1. Introduction

The proximal tubule of the kidney plays a fundamental role in maintaining body salt, fluid, and glucose homeostasis as well as acid and base balance by reabsorbing approximately 65% of filtered sodium (Na^+^), 99% of filtered glucose, and 80–90% of filtered bicarbonate (HCO_3_^-^) [1,2,3,4,5,6]. The proximal tubule of the kidney expresses abundant Na^+^ antiporters and Na^+^-dependent glucose or HCO_3_^-^ cotransporters, as well as Na^+^/K^+^-ATPase, that contribute to maintaining physiological proximal tubule function [4,5,6,7,8,9,10]. On the apical membrane side, NHE3 (Na^+^/H^+^ exchanger 3), an ≈ 85-kDa protein encoded by the *SLC9A3* gene, acts as a powerful driving force for directly and indirectly reabsorbing > 50% of the filtered Na^+^ load in the proximal tubules [10,11,12,13]. The sodium and glucose cotransporter 2 (SGLT2), an ≈ 75-kDa protein encoded by the *SLC5A2* (solute carrier family 5) gene, directly reabsorbs ~90% of filtered glucose in early S1 and S2 proximal tubules [5,11]. The enzyme carbonic anhydrase II converts CO_2_ and H_2_O to H^+^ and HCO_3_^-^, and the latter is then reabsorbed into the blood by the basolateral Na^+^/HCO_3_^-^ cotransporter NBCe1-A [1,2,6,14]. The key action of NHE3 is to secrete the H^+^ ion into the tubular lumen in exchange for the entry of Na^+^ into the proximal tubule cells [3,4,10,13], whereas Na^+^/K^+^-ATPase powers the transport of intracellular Na^+^ ions across the basolateral membranes into the interstitial fluid compartment and then to the blood [7,8]. Together, NHE3, SGLT2, carbonic anhydrase II, Na^+^/HCO_3_^-^ cotransporter, and Na^+^/K^+^-ATPase in the proximal tubules coordinate to regulate Na^+^, glucose, and fluid homeostasis, blood pressure, and acid and base balance in the kidney. 

In the proximal tubule, the expression and/or activity of NHE3, SGLT2, Na^+^/K^+^-ATPase, and Na^+^/HCO_3_^-^ cotransporter are regulated by vasoactive endocrine and local paracrine factors. Angiotensin II (Ang II), the key active peptide of the renin-angiotensin system (RAS), is one of the most important humoral factors in regulating these Na^+^ transporters or cotransporters in the proximal tubules [12,13,14,15]. Extracellular (endocrine or paracrine) Ang II reportedly induces biphasic responses in Na^+^ transport and Na^+^/HCO_3_^-^ cotransport in cultured proximal tubule cells, in vitro isolated proximal tubule perfusion, or in vivo free-flow micropuncture studies with physiological levels (femtomole to picomoles) of this peptide stimulating and pharmacological levels (nanomole to micromoles) inhibiting Na^+^ and Na^+^/HCO_3_^-^ cotransport [16,17,18,19]. These effects of Ang II are mediated by two key subtypes of G protein-coupled receptors, AT_1_ (AT_1a_) and AT_2_ [20,21,22,23,24,25]. In the proximal tubules of the kidney, Ang II binds to cell surface AT_1_ (AT_1a_) receptors in apical and basolateral membranes to activate Gq protein-mediated downstream signaling transduction to generate inositol triphosphate (IP_3_) and diacylglycerol (DAG), which increase intracellular Ca^2+^ and induce protein kinase C activation [20,21,22,23,24,25]. Ang II also binds to AT_1_ (AT_1a_) receptors to induce Gs protein-mediated activation of adenylyl cyclase to generate cAMP [24,25,26,27]. AT_1_ (AT_1a_) receptors play a key role in regulating Na^+^ transport and Na^+^/HCO_3_^-^ cotransport in the proximal tubules and basal blood pressure homeostasis [22,24,25]. By contrast, AT_2_ receptors act through G protein-coupled activation of the bradykinin/nitric oxide/cyclic guanosine 3′,5′ monophosphate signaling pathways to counteract the Na^+^-retaining action of AT_1_ (AT_1a_) receptors [21,23,27]. Both AT_1_ (AT_1a_) and AT_2_ receptors in the proximal tubules contribute to basal blood pressure and body salt and fluid balance by regulating Na^+^ reabsorption from the apical membranes and Na^+^/HCO_3_^-^ cotransport from the basolateral membranes.

We and others have previously reported that in addition to activating classic G protein-coupled signaling, the binding of Ang II to apical as well as basolateral membrane (plasma) AT_1_ (AT_1a_) receptors also simultaneously triggers the AT_1_ (AT_1a_) receptor-mediated internalization or uptake of circulating and intratubular paracrine Ang II into proximal tubule cells in vitro and in vivo [28,29,30,31]. Several proof-of-concept studies have demonstrated that internalized or intracellularly administered Ang II may bind to and stimulate intracellular AT_1_ (AT_1a_) to induce important intracellular responses [32,33,34,35,36,37]. However, whether intracellular Ang II plays a biological and/or physiological role in the expression of NHE3, SGLT2, and Na^+^/HCO_3_^-^ cotransporters in proximal tubule cells and downstream signaling mechanisms has not been well studied. In the present study, we tested the hypothesis that extracellular Ang II is taken up by the proximal tubules of the kidney by an AT_1_ (AT_1a_) receptor-dependent mechanism and that overexpression of an intracellular Ang II fusion protein (ECFP/Ang II) in mouse proximal tubule cells (mPTC) stimulates the expression of Na^+^/H^+^ exchanger 3 (NHE3), sodium and glucose cotransporter 2 (SGLT2), and Na^+^/HCO_3_^-^ cotransporter by AT_1a_/MAPK/ERK1/2/NF-kB signaling mechanisms.

## 2. Methods and Materials

The authors will make all methods and materials, including adenoviral constructs of intracellular Ang II fusion protein, ECFP/Ang II, the sodium and glucose cotransporter 2 (SGLT2) promoter, in vitro transfection protocols, in vivo overexpression of ECFP/Ang II selectively in the proximal tubules of the kidney, reagents, antibodies for Western blot analysis, and all other supporting data, available to other researchers upon request. 

### 2.1. Animals

In this study, wild-type littermates (WT) and global AT_1a_ receptor-deficient (*Agtr1a^-/-^*) mice were used to test our hypothesis. Global *Agtr1a^-/-^* mice were initially purchased from Jackson Labs and are currently maintained in this laboratory [22,31]. The animal studies using [^125^I]-Ang II were previously approved by the Institutional Animal Care and Use Committees of Henry Ford Health System and the University of Mississippi Medical Center, whereas animal studies with the uptake of Alexa 488^®^-labeled Ang II or with overexpression of intracellular cyan fluorescent Ang II fusion protein, ECFP/Ang II, in the proximal tubules of the kidney were approved by the Institutional Animal Care and Use Committees of the University of Mississippi Medical Center and Tulane University School of Medicine, respectively.

### 2.2. Mouse Proximal Tubule Cells with or without the Expression of AT_1_ (AT_1a_) Receptors

Wild-type (WT) mouse proximal tubule cell (mPCT) and *Agtr1a^-/-^* mouse proximal tubule cell lines, as generated from adult male C57BL/6J and *Agtr1a^-/-^* mice, were kindly provided by Dr. Ulrich Hopfer of Case Western Reserve University for the in vitro studies [37,38]. 

### 2.3. AT_1_ (AT_1a_) Receptor-Mediated Uptake of Systemically Infused [^125^I]-Ang II by the Kidney Proximal Tubules in Mice

As a proof-of-concept experiment that circulating, systemic, or paracrine Ang II is taken up by mPCT cells to act as an intracellular peptide in the proximal tubules of the kidney, a radioligand [^125^I]-Ang II was intravenously infused for two hours as we and others described previously [31,39]. Briefly, 2 groups (*n* = 8 per group) of adult male Sprague-Dawley rats (WT) and global AT_1a_ receptor-deficient mice (*Agtr1a^-/-^*) were anesthetized with pentobarbital sodium (50 mg/kg, i.p.), and the left jugular vein was catheterized with a PE-10 catheter for intravenous infusion of [^125^I]-Ang II (specific activity 2176 Ci/mmol, ~10 μCi/25 g body wt) for two hours. Generally, steady-state plasma levels of [^125^I]-Ang II infusion were reached by 30 min, whereas steady-state tissue levels of [^125^I]-Ang II in the kidney proximal tubules were reached within 60 min, as described previously [31,39]. At the end of the experiment, all animals were perfused with acidic buffered saline for 5 min to dissociate cell surface receptor-bound [^125^I]-Ang II from cell membranes and wash out all extracellular [^125^I]-Ang II. The kidneys were removed, [^125^I]-Ang II activity was measured using a gamma counter, and kidneys were sectioned and exposed to X-ray films for 3 days for localization of the intracellular uptake of [^125^I]-Ang II in the kidney, as we described previously [31,39].

### 2.4. AT_1_ (AT_1a_) Receptor-Mediated Uptake of Systemically Infused Alexa 488^®^-Ang II by the Kidney Proximal Tubules in Mice

To further confirm whether circulating, systemic, or paracrine Ang II is taken up by mPCT cells in vivo, 2 groups (*n* = 8 per group) of WT and global AT_1a_ receptor-deficient mice (*Agtr1a^-/-^*) were anesthetized with pentobarbital sodium (50 mg/kg, i.p.), and the left jugular vein was catheterized with a PE-10 catheter for intravenous infusion of fluorescent Alexa 488^®^-Ang II (Invitrogen, Waltham, MA, USA, 10 ng/min, Cat: A13439; Lot: 828266) for two hours. As described for [^125^I]-Ang II, steady-state plasma and proximal tubule uptake levels of Alexa 488^®^-Ang II were reached by 30 and 60 min, respectively. At the end of the experiment, mice were perfused with acidic buffered saline for 5 min to dissociate cell surface receptor-bound Alexa 488^®^-Ang II from cell membranes and wash out all extracellular Alexa 488^®^-Ang II. The kidneys were removed, sectioned, and visualized for localization of the intracellular uptake of Alexa 488^®^-Ang II in the kidney using an A1-HD25 Nikon Confocal Laser Scanning Microscopy System with a 488 filter [37].

### 2.5. Construction of Proximal Tubule Cell-Specific Intracellular Ang II Fusion Protein, Ad-Sglt2-ECFP/Ang II 

It is difficult to completely separate the effects of intracellular Ang II mediated by cytoplasmic receptors from those of extracellular Ang II mediated by cell surface receptors in a practical manner. Therefore, we constructed a proximal tubule cell-specific, enhanced cyan fluorescent protein (EFCP)-tagged intracellular Ang II fusion protein for in vitro expression in mPCT cells or for in vivo expression in the proximal tubules of the mouse kidneys, as described by us and others previously [36,37,40]. The construct for this ECFP-tagged intracellular Ang II fusion protein, ECFP/Ang II, was kindly provided by Dr. Julia Cook of the Ochsner Clinic Foundation [36,40]. A proximal tubule-specific SGLT2 promoter, pGEM-sglt2-5pr-mut, was kindly provided by Drs. Isabelle Rubera and Michel Tauc of the Université Côte d’Azur, France [41]. An adenoviral construct encoding ECFP/Ang II and the SGLT2 promoter, Ad-*Sglt2-ECFP/Ang II*, was custom-designed, amplified, and purified by Vector BioLab for in vitro expression in mPCT cells or for in vivo expression in the proximal tubules of the mouse kidneys, as we described previously [22,37]. 

### 2.6. Adenovirus-Mediated Overexpression of Ad-Sglt2-ECFP/Ang II Selectively in the Proximal Tubules of WT Mouse Kidneys

Two groups (*n* = 8 each) of adult male WT littermates or global *Agtr1a^-/-^* mice were infected without (time controls) or with adenovirus-mediated, proximal tubule-specific expression of Ad-*Sglt2-ECFP/Ang II* in the kidney for 2 weeks [22,37]. Briefly, animals were anesthetized with sodium pentobarbital (50 mg/kg, i.p.), and their renal arteries were exposed by a left and right flank incision, followed by temporarily clamping with a fine vessel clip, briefly interrupting blood flow to the kidney for 5 min. Ad-*Sglt2-ECFP/Ang II* was diluted 1:5 in sterilized phosphate-buffered saline and directly injected into the superficial cortex evenly in six different locations (20 μL each) [22,37]. The control groups of animals received only sham injections of saline instead. Renal blood flow was reestablished 5 min after injection of Ad-*Sglt2-ECFP/Ang II* or saline. Animals were allowed to recover from the surgery and express Ad-*Sglt2-ECFP/Ang II* for two weeks. At the end of the experiment, the in vivo expression of Ad-*Sglt2-ECFP/Ang II* in the proximal tubules of the kidney was visualized by an A1-HD25 Nikon Confocal Laser Scanning Microscopy System using a CFP filter [22,37]. These animal studies using an adenoviral construct were approved by the Institutional Recombinant DNA and Biosafety Committees of the University of Mississippi Medical Center and Tulane University School of Medicine, respectively. 

### 2.7. Adenovirus-Mediated Expression of Ad-Sglt2-ECFP/Ang II in mPCT Cells with or without AT_1a_ Receptors

To determine whether intracellular Ang II induces important biological responses, Ad-*Sglt2-ECFP/Ang II* was expressed in mPCT cells (abbreviation: ECFP/Ang II). Briefly, WT and *Agtr1a^-/-^* mPCT cells were split into 6-well plates and cultured to 80% confluency, and then transfected with Ad-*Sglt2-ECFP/Ang II* (~4 µg/well) for 48 h using a standard transfection protocol as previously described [22,37]. In another series of experiments, *Agtr1a^-/-^* mPCT cells were transfected concurrently with Ad-*Sglt2-ECFP/Ang II* and a full-length mouse *Agtr1a* cDNA ORF clone (NM_177322, OriGene, Rockville, MD, USA) to rescue the AT_1a_ receptor expression in *Agtr1a^-/-^* mPCT cells before they were treated with the inhibitors.

### 2.8. Pharmacological Inhibitors of AT_1_ (AT_1a_) or AT_2_ Receptor-Mediated, MAP Kinase ERK1/2- and NF-κB-Dependent Signal Pathways

To determine the potential signaling mechanisms involved in Ad-*Sglt2-ECFP/Ang II*-induced biological responses, WT and *Agtr1a^-/-^* mPCT cells expressing Ad-*Sglt2-ECFP/Ang II* were concurrently treated with the AT_1_ receptor antagonist losartan (10 µM; Tocris, Minneapolis, MN, USA), the AT_2_ receptor antagonist PD 123319 (10 µM; Tocris, Minneapolis, MN, USA), the MEK1/MEK2 kinase inhibitor U0126 (1 µM; Tocris, Minneapolis, MN, USA), the MEK inhibitor PD 980659 (1 µM; Tocris, Minneapolis, MN, USA), the NF-κB activation inhibitor RO 106–9920 (10 µM; Tocris, Minneapolis, MN, USA), and the p38 MAP kinase inhibitor SB202196 (10 µM; MCE, Belleville, NJ, USA).

### 2.9. Antibodies Targeting Signaling and Sodium Transporter or Cotransporter Proteins

The following antibodies were used in the present study to determine the expression of key Na^+^ transporter or cotransporter proteins or signaling pathway proteins. A mouse monoclonal antibody targeting a short amino acid sequence containing dually phosphorylated Thr 202 and Tyr 204 (pT202/pY204) of MAP kinases ERK1/2 of rat origin (sc-136521) and a rabbit polyclonal antibody targeting a short amino acid sequence containing phosphorylated Ser 276 of the NF-κB, p65 subunit of human origin (sc-101749) were purchased from Santa Cruz Biotechnology (Santa Cruz, CA, USA). A rabbit polyclonal antibody targeting a synthetic peptide (KLH-coupled) in the C terminus of the rat MAP kinase ERK 1/2 (no. 9102) was purchased from Cell Signaling. A rabbit anti-Na^+^/HCO_3_ cotransporter polyclonal antibody (AB3212), a monoclonal antibody targeting a fusion protein containing the C-terminal 131 amino acids of rabbit NHE3 (no. MAB3136), and a mouse monoclonal antibody targeting a synthetic peptide corresponding to human NF-κB, p65 subunit, anti-NF-κB, p65 subunit clone 12H11 (no. MAB3026) were purchased from Millipore, respectively. 

### 2.10. Western Blot Analysis 

At the end of the treatments, protein samples were extracted from all mPCT cells for Western blot analysis of the expressions. Protein concentrations were measured using a bicinchoninic acid protein assay kit (Pierce, Appleton, WI, USA) and GraphPad Prism 9.0 [15,37]. Briefly, 10 to 20 μg each of protein samples were electrophoretically separated on 8–16% Tris-glycine gels at 120 volts for 1.5–2.0 h and transferred to Millipore Immobilon-P membranes using a Bio-Rad Trans-Blot Semi-Dry system (25 V, 0.12 A, 1.5 h). All membranes were blotted overnight at 4 °C with 5% nonfat dry milk and incubated with a specific antibody as described above for 3 h at room temperature. The same membranes were then treated with a stripping buffer (Pierce) for 20 min, blotted with 5% nonfat dry milk, and re-probed with a mouse anti-β-actin monoclonal antibody at 1:2000 (Sigma-Aldrich, St. Louis, MO, USA) [15,37]. At least 6 wells/samples from untreated (controls) and inhibitor-treated wild-type or *Agtr1a*^-/-^ mPCT cells were included in all in vitro cell culture studies. Western blot signals for each signaling or transporter protein were detected using enhanced chemiluminescence (Amersham, Amersham, UK) and analyzed using a BIO-RAD ChemiDoc ^TM^ MP Imaging System and GraphPad Prism 9.5.1.

### 2.11. Statistical Analysis 

All data are presented as means ± SE. For semi-quantitative comparisons, all Western blot responses were expressed as a ratio of a specific protein to β-actin expression. Statistical comparisons were analyzed first using one-way ANOVA to compare the difference in the same response between control and treated WT and *Agtr1a^-/-^* mPCT cells and between WT and *Agtr1a^-/-^* mice. If the *p* value was less than 0.05, a post-hoc Newman-Keul multiple comparison test was then used to compare two different group means. The significance of differences was set at *p* < 0.05. 

## 3. Results

### 3.1. AT_1_ (AT_1a_) Receptor-Mediated Uptake of [^125^I]-Labeled Ang II in the Rat and Mouse Kidneys 

To confirm whether the circulating endocrine Ang II is taken up by the rat and mouse kidneys, especially in the proximal tubules, the intracellular uptake of [^125^I]-labeled Ang II in the kidney was directly compared in rats treated with or without losartan pretreatment and in wild-type and whole-body *Agtr1a^-/-^* mice, respectively (Figure 1). In response to systemic infusion of the same doses of [^125^I]-labeled Ang II without pretreatment with losartan to block all AT_1_ receptors, the rat kidney markedly accumulated systemically infused [^125^I]-labeled Ang II primarily in the superficial cortex, corresponding to the glomeruli and the proximal tubules (Figure 1A). By contrast, chronic pretreatment of the rats with losartan for one week to block AT_1_ receptors (20 mg/kg/day, p.o.) nearly completely blocked the uptake of [^125^I]-labeled Ang II in the rat kidney (Figure 1B). The intracellular level of [^125^I]-labeled Ang II uptake in the kidney was more than 6.5-fold higher in the rats not pretreated with losartan than in the rats pretreated with losartan (*p* < 0.01 vs. ^125^I-labeled Ang II) (Figure 1C). In wild-type mice, the intracellular uptake of systemically infused [^125^I]-Ang II was about 8 times higher in wild-type mice (Figure 1D) than in global AT_1a_ receptor-deficient mice, *Agtr1a^-/-^* mice (*p* < 0.01) (Figure 1E,F).

### 3.2. AT_1a_ Receptor-Mediated Uptake of Alexa 488-Labeled Ang II in Wild-Type and Agtr1a^-/-^ Mouse Kidneys

To further confirm and visualize whether the circulating Alexa 488^®^-labeled Ang II is taken up by the proximal tubules of mouse kidneys, the intracellular uptake of Alexa 488^®^-labeled Ang II in the kidney was directly compared between wild-type and *Agtr1a^-/-^* mice (Figure 2). In *Agtr1a^-/-^* mice, there was a minimal level of Alexa 488^®^-labeled Ang II uptake by the glomeruli and the proximal tubules, suggesting that deletion of AT_1a_ receptors in mice blocked Alexa 488^®^-labeled Ang II uptake by the kidney (Figure 2A). In wild-type mice, the uptake of Alexa 488^®^-labeled Ang II was very intensive in the proximal tubules, as strong green fluorescence was observed over the proximal tubules in the superficial cortex (*p* < 0.01 vs. *Agtr1a^-/-^* mice) (Figure 2B). However, the glomeruli and the cortical collecting tubules did not significantly take up systemically infused Alexa 488^®^-labeled Ang II, suggesting that this response is a proximal tubule-specific phenomenon (Figure 2B).

### 3.3. Adenovirus-Mediated, Proximal Tubule-Specific Overexpression of Intracellular Ang II Fusion Proteins in Wild-Type and Agtr1a^-/-^ Mouse Kidneys

We then designed and constructed an adenovirus-mediated, proximal tubule-specific, intracellular Ang II fusion protein, Ad-*Sglt2-ECFP/Ang II*, for in vivo expression in wild-type mice, as described in [36,37,41]. Figure 3 shows that control mice showed minimal autofluorescence in the proximal tubules of the kidney (Figure 3A, C). By contrast, intensive cyan fluorescence (blue green) representing the expressed Ad-*Sglt2-ECFP/Ang II* was observed in the proximal tubules of the mouse kidney (*p* < 0.01 vs. control) (Figure 3B). No significant expression of Ad-*Sglt2-ECFP/Ang II* was visualized in the glomerulus (Figure 3B). These results demonstrated a proof-of-concept approach to expressing an intracellular Ang II fusion protein in the proximal tubules of the kidney using the sodium and glucose cotransporter 2 promoter (Sglt2) to study the effects and mechanisms of intracellular Ang II-induced expression of sodium transporters or cotransporters in mouse proximal tubule cells (see below).

### 3.4. Role of AT_1_ (AT_1a_) and AT_2_ Receptors in Mediating Intracellular Ang II Fusion Protein-Induced NHE3 Expression in mPCT Cells

Figure 4 shows that the expression of an intracellular Ang II fusion protein (ECFP/Ang II) is biologically active in significantly inducing the expression of NHE3 by nearly 3-fold, one of the most important sodium transporters responsible for reabsorbing over 50% of the filtered sodium load by the glomeruli in WT mPCT cells (*p* < 0.01 vs. control) (Figure 4A). The effect of ECFP/Ang II on NHE3 expression was markedly attenuated by losartan, an AT_1_ receptor blocker (*p* < 0.01 vs. ECFP/Ang II). In *Agtr1a^-/-^* mPCT cells, however, the expression of ECFP/Ang II had a markedly smaller effect on NHE3 expression (n.s. vs. control) (Figure 4B). Both the AT_1_ receptor blocker losartan and the AT_2_ receptor blocker PD123319 had no significant effects on NHE3 expression in *Agtr1a^-/-^* mPCT cells. These data confirm that proximal tubule-specific expression of intracellular Ang II fusion proteins induces NHE3 expression in wild-type but not *Agtr1a^-/-^* mPCT cells.

### 3.5. Roles of the MAP Kinase ERK1/2, p38 MAPK, or NF-кB Signaling Pathways in Mediating Intracellular Ang II Fusion Protein-Induced NHE3 Expression in mPCT Cells

Figure 5 shows several potential downstream signaling mechanisms underlying AT_1_ (AT_1a_) receptor-mediated ECFP/Ang II-induced NHE3 expression in mPCT cells. In WT mPCT cells, the stimulated effect of ECFP/Ang II on NHE3 expression was markedly attenuated by U0126, a highly selective inhibitor of the MAP kinases MEK1 and MEK2 (*p* < 0.01 vs. control) (Figure 5A). Concurrent treatment of WT mPCT cells expressing ECFP/Ang II with Ro 106-9920, a cell-permeable inhibitor of NF-κB activation, also blocked the effect of ECFP/Ang II on NHE3 expression (*p* < 0.01 vs. control) (Figure 5A). However, concurrent treatment of WT mPCT cells expressing ECFP/Ang II with either PD980659 or SB202196, a selective p38 MAP kinase inhibitor, had no significant effect (n.s.) (Figure 5A). By comparison, U0126, Ro 106-9920, PD980659, and SB202196 did not have significant effects on ECFP/Ang II-induced NHE3 expression in *Agtr1a^-/-^* mPCT cells (Figure 5B).

### 3.6. Role of AT_1_ (AT_1a_) and AT_2_ Receptors in Mediating Intracellular Ang II Fusion Protein-Induced Na^+^/HCO_3_^-^ Expression in mPCT Cells

The Na^+^/HCO_3_^-^ cotransporter plays a key role in proximal tubular reabsorption of HCO_3_^-^ and in maintaining intracellular pH. Figure 6 shows that the expression of intracellular ECFP/Ang II also significantly increased the expression of Na^+^/HCO_3_^-^ cotransporter in WT mPCT cells (Figure 6A; *p* < 0.01 vs. control). This stimulatory response was markedly attenuated to a level below control by losartan, but PD123319 also attenuated the effect of ECFP/Ang II on Na^+^/HCO_3_^-^ expression (*p* < 0.01 vs. ECFP/Ang II alone). The potential signaling mechanisms involved in ECFP/Ang II-induced Na^+^/HCO_3_^-^ expression are shown in Figure 6B. The MAP kinase MEK1/MEK2 inhibitor U0126, the ERK1/2 inhibitor PD980659, as well as the inhibitor of NF-κB activation Ro-106 9920, all significantly attenuated ECFP/Ang II-induced Na^+^/HCO_3_^-^ expression (Figure 6B; *p* < 0.01 vs. ECFP/Ang II). However, the p38 MAP kinase inhibitor SB202196 had no effect on ECFP/Ang II-induced Na^+^/HCO_3_^-^ expression, suggesting that p38 MAP kinase may not be involved in this response.

### 3.7. Roles of AT_1_ (AT_1a_) and AT_2_ Receptors in Mediating Intracellular Ang II Fusion Protein-Induced Sglt2 Expression in mPCT Cells

The sodium and glucose cotransporter 2 (Sglt2) is responsible for reabsorbing nearly all filtered glucose from the glomeruli and a small fraction of sodium in the proximal tubules of the kidney. Figure 7 shows that the expression of an intracellular Ang II fusion protein (ECFP/Ang II) significantly induced the expression of the sodium and glucose cotransporter 2 (Sglt2) by > 2-fold in WT mPCT cells (*p* < 0.01 vs. control) (Figure 7A). The effect of ECFP/Ang II on Sglt2 expression was markedly attenuated by losartan or PD123319 (*p* < 0.01 vs. ECFP/Ang II), suggesting that both AT_1_ and AT_2_ receptors are involved. In *Agtr1a^-/-^* mPCT cells, ECFP/Ang II alone did not induce Sglt2 expression, but Sglt2 expression was increased significantly in the presence of losartan (*p* < 0.01 vs. ECFP/Ang II) and decreased significantly in the presence of PD123319 (*p* < 0.01 vs. ECFP/Ang II + Los). These data suggest that both AT_1_ and AT_2_ receptors are involved in ECFP/Ang II-induced Sglt2 expression in mPCT cells (Figure 7A, B).

### 3.8. Roles of AT_1_ (AT_1a_) and AT_2_ Receptors in Mediating Intracellular Ang II Fusion Protein-Induced NF-кB, p65 Expression in mPCT Cells

Nuclear factor NF-kappaB p65 subunit (NF-кB, p65) is an important transcription factor encoded by the RELA gene and plays a key role in NF-κB heterodimer formation, nuclear translocation, and activation [42]. Figure 8 shows that the expression of ECFP/Ang II in WT mPCT cells markedly increased NF-кB, p65 protein expression, and interestingly, this stimulatory response was blocked by concurrent treatment with losartan and PD123319 (Figure 8A), again suggesting that both AT_1_ and AT_2_ receptors are involved. This was supported by the lack of the NF-кB, p65 response to ECFP/Ang II expression in *Agtr1a^-/-^* mPCT cells (Figure 8B), while concurrent treatment with losartan or PD123319 increased NF-кB, p65 expression in *Agtr1a^-/-^* mPCT cells (Figure 8B). Further experiments showed that the MAP kinase MEK1/MEK2 inhibitor U0126, the MEK inhibitor PD980659, as well as the inhibitor of NF-κB activation Ro 106-9920, all markedly attenuated ECFP/Ang II-induced NF-кB, p65 expression (Figure 8C). However, the p38 MAP kinase inhibitor SB202196 again had no effect on ECFP/Ang II-induced NF-кB, p65 expression, suggesting that p38 MAP kinase may not be involved in this response (Figure 8C).

### 3.9. Comparisons of Extracellular Ang II and Intracellular ECFP/Ang II-Induced MAP Kinases ERK1/2 Activation in WT and Agtr1a^-/-^ mPCT Cells

It is well known that extracellular Ang II can induce the activation of MAP kinases ERK1/2 through cell surface Ang II receptors in different cell types and tissues [20,24,25], but whether intracellular Ang II can also activate MAP kinases ERK1/2 through intracellular Ang II receptors remains poorly understood. Figure 9 shows that, as expected, extracellular Ang II induced MAP kinases ERK1/2 activation in WT mPCT cells in time- and concentration-dependent manners, with a maximal response at 5 min of stimulation by 10 nM Ang II (Figure 9A). Additionally, it is expected that extracellular Ang II-induced activation of the MAP kinases ERK1/2 in WT mPCT cells was markedly blocked by concurrent treatment with losartan or PD123319 (Figure 9B). Interestingly, the expression of intracellular ECFP/Ang II also markedly increased MAP kinases ERK1/2 expression in WT mPCT cells, and this response was blocked by losartan but not by PD123319 (Figure 9C), which was different from extracellular Ang II (Figure 9B). In *Agtr1a^-/-^* mPCT cells, the expression of intracellular ECFP/Ang II did not increase MAP kinases ERK1/2 expression as in WT mPCT cells (Figure 10A), whereas the knock-in of AT_1a_ receptors in *Agtr1a^-/-^* mPCT cells successfully rescued the stimulatory effect of ECFP/Ang II expression on MAP kinases ERK1/2 activation (Figure 10B). Taken together, these data provide proof-of-concept evidence that, similar to extracellular Ang II, intracellular Ang II may also activate the MAP kinases ERK1/2 to induce important biological and physiological effects and important signaling responses in mPCT cells.

### 3.10. Knocking in a Full-Length Wild-Type Mouse AT_1a_ Receptor in Agtr1a^-/-^ mPCT Cells Rescues the Intracellular ECFP/Ang II-Induced Activation of the MAP Kinase ERK1/2 Signaling Response

In *Agtr1a*^-/-^ mPCT cells, the expression of intracellular ECFP/Ang II did not significantly increase the MAP kinases ERK1/2 activation, but concurrent treatment with losartan, not with PD123319, significantly increased p-ERK1/2 expression in *Agtr1a*^-/-^ mPCT cells, suggesting a dominant role of AT_1_ receptors (Figure 10A). However, the knocking in of a full-length wild-type mouse AT_1a_ receptor in *Agtr1a*^-/-^ mPCT cells completely rescued the p-ERK1/2 signaling response to ECFP/Ang II expression, which was again blocked by losartan but not by PD123319 (Figure 10B).

## 4. Discussion

The present study demonstrates three key findings to support the proof-of-concept hypothesis of an important role for intracellular Ang II and its receptors in mPTC cells and the proximal tubules of the kidney. First, we found that [^125^I]-labeled Ang II or Alexa 488^®^-labeled Ang II, systemically infused as an extracellular form of Ang II (circulating and paracrine), was taken up by the proximal tubules of the rat and mouse kidneys by the AT_1_ (AT_1a_) receptor-mediated uptake mechanism. This conclusion is supported by the findings that proximal tubular uptake of systemically infused [^125^I]-labeled Ang II or Alexa 488^®^-labeled Ang II in the kidney was blocked by pretreatment with losartan, an AT_1_ receptor blocker, or in mutant mice with whole-body deletion of AT_1a_ receptors (*Agtr1a^-/-^*). Our results are also consistent with a number of previous studies in which systemically infused Ang II was accumulated by the kidneys of rats [28,30], pigs [29], and mice and by the AT_1_ receptor-mediated mechanism [31]. However, the results of the present study differ significantly from those of the above-mentioned studies in that we could visualize the intra-kidney localization of [^125^I]-labeled Ang II or Alexa 488^®^-labeled Ang II primarily in the outer cortex but not in the entire kidney, which is consistent with a primary proximal tubular localization. Indeed, vat Kats previously infused [^125^I]-labeled Ang II in the pigs and analyzed the subcellular localization of internalized [^125^I]-labeled Ang II in different intracellular organelles of the kidney [29,39]. These investigators identified [^125^I]-labeled Ang II in cytosol-, lysosome-, mitochondrial-, and nuclei-enriched fractions, respectively [29,39]. They further found that the AT_1_ receptor blocker eprosartan greatly decreased [^125^I]-labeled Ang II levels in these organelles, suggesting that the AT_1_ receptor mediates the [^125^I]-labeled Ang II uptake by the pig kidneys. Using a different approach, we infused Ang II in the rats for two weeks to induce Ang II-dependent hypertension and isolated light and heavy endosomal fractions from the kidney cortex, primarily the proximal tubules, to measure Ang II and AT_1_ (AT_1a_) receptor levels [30]. We found that Ang II levels in the kidney cortical endosomes were about 10-fold higher than in the plasma, where internalized Ang II was colocalized with AT_1_ (AT_1a_) receptors [30]. In rats concurrently treated with candesartan, an AT_1_ receptor antagonist tightly bound to cell surface AT_1_ receptors, the kidney cortical endosomal uptake of Ang II was blocked, again suggesting an AT_1_ receptor-mediated mechanism [30].

Since the kidney proximal tubules take up or accumulate abundant extracellular Ang II by AT_1_ (AT_1a_) receptor-mediated mechanisms in different intracellular compartments (or organelles) in addition to the endosome/lysosome pathway, an interesting question arises with respect to whether internalized Ang II and AT_1_ (AT_1a_) receptors are biologically active or physiologically relevant in the proximal tubules. Previous studies have shown that extracellular (circulating, endocrine, and paracrine) Ang II binds and activates cell surface AT_1_ and AT_2_ receptors to regulate the expression and actions of NHE3 [15,23,25,35,42,43], SGLT2 [44,45], and Na^+^/HCO_3_^-^ cotransporters in the proximal tubules of the kidney [14,18,19]. However, little is known regarding whether these responses are mediated only by Ang II acting on cell surface Ang II receptors or in part by internalized Ang II acting as an intracellular Ang II to induce the expression of NHE3, SGLT2, and Na^+^/HCO_3_^-^ cotransporters in the proximal tubules. To test this hypothesis, the present study overexpressed an intracellular Ang II fusion protein, ECFP/Ang II, selectively in mPCT cells as a novel approach to determining whether intracellular Ang II induces the expression of key Na^+^ transporters or Na^+^ cotransporters. Indeed, the present study was able to demonstrate that the expression of an intracellular Ang II fusion protein significantly induced the expression of these key Na^+^ transporters or cotransporters in mPCT cells, therefore supporting the proof-of-concept hypothesis that intracellular Ang II and AT_1_ (AT_1a_) receptors may be biologically or physiologically active to regulate proximal tubule Na^+^ reabsorption in the kidney. However, we do recognize that one cannot equate the expression of this intracellular Ang II fusion protein to endogenous intracellular Ang II or the internalized Ang II-AT_1_ receptor complex. Indeed, other alternative approaches have been used by others to determine whether intracellular Ang II plays important biological or physiological roles in other cells. For example, Haller et al. microinjected Ang II directly into cultured rat vascular smooth muscle cells (VSMCs) and demonstrated that microinjected Ang II significantly increased cytosolic and nuclear calcium mobilization in VSMCs [46]. In cardiomyocytes, De Mello showed that dialysis of Ang II directly into the cells significantly stimulated the inward calcium current, implying an intracellular action of Ang II on cellular communication [34]. In cardiac fibroblasts, Tadevosyan et al. demonstrated that intracellular Ang II released by photolysis of a membrane-permeable caged Ang II analog induced IP_3_R-dependent nucleoplasmic Ca^2+^mobilization and regulated fibroblast proliferation and collagen-1A1 mRNA expression [47]. Alternatively, a novel molecular biology approach was developed by Cook et al. [36,40]. Cook et al. designed an intracellular Ang II fusion protein fused to enhanced cyan fluorescent protein (ECFP), ECFP/Ang II, whose expression is confined intracellularly and biologically active without being secreted into the extracellular compartment to induce endocrine and local paracrine responses [36,40]. Overexpression of this intracellular Ang II fusion protein induced the expression of platelet-derived growth factor (PDGF) in rat hepatoma cells and activated cAMP response element-binding protein (CREB) activity in CHO-K1 and COS-7 cells [36,40], whereas global expression of ECFP/Ang II in mice led to elevated blood pressure and induced renal thrombotic microangiopathy [48]. Furthermore, we have recently used a proximal tubule-specific sodium and glucose cotransporter 2 (Sglt2) promoter to drive the overexpression of ECFP/Ang II selectively in the proximal tubules of the rat and mouse kidneys [22,37]. We demonstrated that proximal tubule-specific overexpression of ECFP/Ang II significantly increased proximal tubule Na^+^ reabsorption, induced an antinatriuretic response, and elevated blood pressure, which were blocked by concurrent treatment with losartan or in global *Agtr1a^-/-^* mice [22,37]. The results of the current study are thus consistent with these previously published studies.

However, the signaling mechanisms involved in increasing proximal tubule Na^+^ reabsorption and blood pressure by overexpressing ECFP/Ang II selectively in the proximal tubules in vivo remain poorly understood. To exclude the influences of other extracellular humoral factors from other kidney cells, we believe that the use of signaling pathway-specific inhibitors and cultured mPCT cells with or without the expression of an intracellular Ang II fusion protein is an ideal approach to elucidate the signaling mechanisms underlying ECFP/Ang II-induced expression of NHE3, SGLT2, and Na^+^/HCO_3_^-^ cotransporters in the proximal tubules [37,49]. It is well recognized that extracellular Ang II, by activating cell surface AT_1_ (AT_1a_) or AT_2_ receptors, induces cellular growth and proliferative or transcriptional responses in many other cell types by MAP kinases ERK1/2 and/or NF-кB signaling [15,20,24,25]. We and others also showed in previous studies that extracellular Ang II induced the activation of the MAP kinases ERK1/2 and NF-кB signaling and the expression of NHE3 and angiotensinogen expression in dose- and time-dependent manners [15,18,25,37]. However, the present study clearly demonstrated that AT_1_ (AT_1a_) receptor-mediated activation of MAP kinases ERK1/2 and nuclear factor-кB signaling pathways is also involved in intracellular ECFP/Ang II-induced expression of NHE3, SGLT2, and Na^+^/HCO_3_^-^ cotransporters in mPCT cells. Indeed, this conclusion is supported by the findings that ECFP/Ang II significantly increased the expression of these Na^+^ transporters or Na^+^ cotransporters, the MAP kinases ERK1/2, and the NF-κB, p65 in mPCT cells, and these responses were blocked by the MEK1/MEK2 kinase inhibitor U0126, the MEK inhibitor PD 980659, and the NF-κB activation inhibitor RO 106–9920, respectively. Nevertheless, the present study does not support a role for p38 MAP kinase, as the p38 MAP kinase inhibitor SB202196 showed no significant effect. Another interesting finding was that AT_2_ receptors may also be involved in intracellular ECFP/Ang II-induced expression of NHE3, SGLT2, and Na^+^/HCO_3_^-^ cotransporters in mPCT cells since the AT_2_ receptor blocker PD123319 also significantly blocked these responses. These findings are somewhat unexpected and in contrast with previous studies showing that AT_2_ receptors mediated extracellular Ang II- and/or Ang III-induced natriuretic responses in the proximal tubules of the rat kidney in part by inducing the internalization of NHE3 from the apical membranes and Ang II-induced inflammatory responses in unilateral ureteral obstruction [23,27,50,51,52]. The present studies raise the possibility that PD123319 may not be a specific blocker of AT_2_ receptors, or at pharmacological concentrations, it may behave similar to losartan to block AT_1_ receptors in mPCT cells in vitro. Further studies are necessary to repeat these experiments by overexpressing intracellular ECFP/Ang II in mPCT cells deficient in AT_2_ receptors or by molecularly silencing AT_2_ receptors in wild-type mPCT cells. Nevertheless, only about 10% of Ang II receptors are AT_2_ receptors in the proximal tubules, and AT_1_ (AT_1a_) receptors still play a very dominant role in overshadowing the roles of AT_2_ receptors in mPCT cells or in the proximal tubules of the kidney. 

In summary, the present study expressed a novel intracellular Ang II fusion protein as a surrogate of an intracellular Ang II in mPCT cells with or without expression of AT_1_ (AT_1a_) receptors and clearly demonstrated that the overexpression of this intracellular Ang II fusion protein selectively in mPTC cells was able to significantly induce the expression of NHE3, SGLT2, and Na^+^/HCO_3_^-^ cotransporters by AT_1_ (AT_1a_) receptor-mediated, MAP kinases ERK1/2- and NF-кB-dependent signaling pathways (Figure 11). These findings are novel in that the current popular paradigm suggests that extracellular Ang II and cell surface AT_1_ (AT_1a_) receptors mediate most, if not all, biological, physiological, and signaling responses, while intracellular Ang II and its receptors play little or no role. Our results therefore provide proof-of-concept evidence supporting the important role of the intracellular Ang II system in the proximal tubules of the kidney, probably independent from the circulating and local tissue paracrine Ang II system. Furthermore, our studies also help explain in vivo physiological and pharmacological studies in rats and mice in which systemic infusion of Ang II for days and weeks continuously elevates blood pressure and induces hypertensive kidney injury despite the fact that cell surface AT_1_ (AT_1a_) receptors are “desensitized” due to Ang II-induced internalization into the endosome/lysosome degradation and recycling pathways. However, we should also recognize the limitations of the present study, namely that the expression of an intracellular Ang II fusion protein as a surrogate approach by no means truly represents endogenous intracellular Ang II in mPCT cells in vitro or in the proximal tubules of the kidney in vivo. Further studies are necessary to develop new molecular biological constructs, in vitro cellular models, pharmacological compounds, and novel proximal tubule-specific mutant mouse models with gain of function or loss of function to further elucidate the roles and mechanisms of intracellular Ang II actions in the kidney.

## Figures and Tables

**Figure 1 cells-12-01492-f001:**
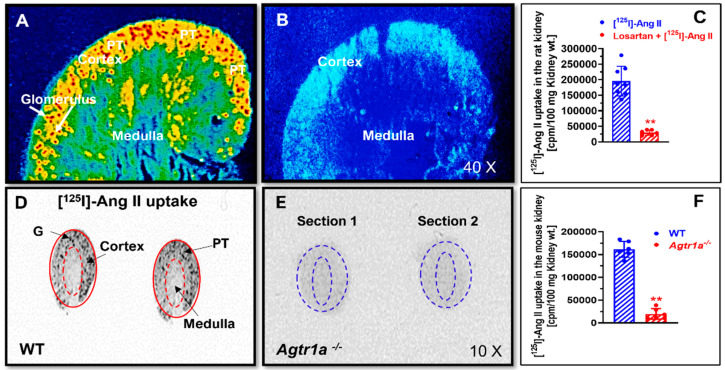
In vivo autoradiographic micrographs showing the intracellular uptake of systemically infused radiolabeled [^125^I]-Ang II in the kidney cortex, primarily in the glomeruli and proximal tubules, in rats and mice. (**A**) Intracellular uptake of [^125^I]-Ang II in the kidney cortex of a representative adult male Sprague-Dawley rat. (**B**) The intracellular uptake of [^125^I]-Ang II in the rat kidney cortex was blocked by pretreatment with the AT_1_ receptor blocker losartan, 20 mg/kg/day, p.o., for 1 week in a representative adult male Sprague-Dawley rat. (**C**) Quantitative data of AT_1_ receptor-mediated intracellular uptake of [^125^I]-Ang II in the rat kidney (*n* = 8). Levels of intracellular [^125^I]-ANG II uptake in the rat kidney are keyed to the color calibration bar, with red representing the highest (H) and blue the background (L) level of uptake. (**D**) Intracellular uptake of [^125^I]-Ang II in the kidney cortex of a representative adult male C57BL/6J mouse (WT). (**E**) The intracellular uptake of [^125^I]-Ang II in the kidney cortex was blocked in a representative global AT_1a_ receptor-deficient mouse (*Agtr1a*^-/-^). (**F**) Quantitative data of AT_1a_ receptor-mediated intracellular uptake of [^125^I]-Ang II in the mouse kidney (*n* = 8). G, glomerulus. PT, proximal tubule. ** *p* < 0.01 vs. control rat or wild-type mouse kidney.

**Figure 2 cells-12-01492-f002:**
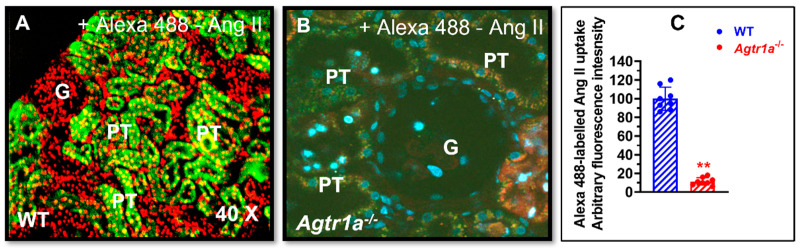
AT_1a_ receptor-mediated intracellular uptake of Alexa fluor 488^®^-labeled Ang II in the proximal tubules of wild-type (WT) and global AT_1a_ receptor-deficient mice (*Agtr1a*^-/-^). (**A**) Intracellular uptake of Alexa fluor 488^®^-labeled Ang II in the proximal tubules of a representative adult male wild-type mouse (WT). (**B**) Intracellular uptake of Alexa fluor 488^®^-labeled Ang II in the proximal tubules was blocked in a representative *Agtr1a*^-/-^ mouse. (**C**) Semiquantitative data on the intracellular uptake of Alexa fluor 488^®^-labeled Ang II in the proximal tubules of WT and *Agtr1a*^-/-^ mice, expressed as arbitrary fluorescence intensity with WT mice at 100. G, glomerulus. PT, proximal tubule. Red represents DAPI-stained nuclei in the cortex, which were converted into RED for a better contrast with Alexa fluor 488^®^-labeled Ang II. ** *p* < 0.01 vs. WT mice. Magnification: ×40.

**Figure 3 cells-12-01492-f003:**
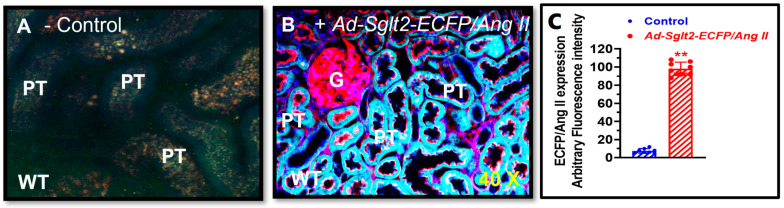
Adenovirus-mediated, proximal tubule-targeting expression of an intracellular cyan fluorescent Ang II fusion protein, Ad-*Sglt2-ECFP/Ang II*, in the proximal tubules of the mouse kidney. (**A**) A representative control mouse kidney without expressing Ad-*Sglt2-ECFP/Ang II* and only showing autofluorescence in the proximal tubules. (**B**) A representative mouse kidney expressing robust Ad-*Sglt2-ECFP/Ang II* in the proximal tubules. (**C**) Semiquantitative data on proximal tubule-specific expression of Ad-*Sglt2-ECFP/Ang II* in the proximal tubules, expressed as arbitrary fluorescence intensity. G, glomerulus. PT, proximal tubule. Cyan, Ad-*Sglt2-ECFP/Ang II*. Blue, nuclei. Red, apical. The basolateral membranes of the tubules as well as the glomeruli were stained by the high-affinity F-actin probe Alexa Fluor 568 phalloidin. ** *p* < 0.01 vs. control mice. Magnification: ×40.

**Figure 4 cells-12-01492-f004:**
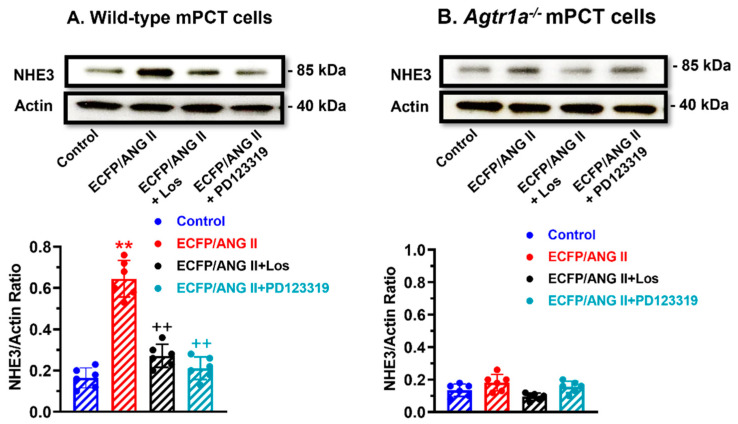
Effects of ECFP/Ang II on Na^+^/H^+^ exchanger 3 (NHE3) expression in wild-type and *Agtr1a*^-/-^ mouse proximal tubule cells (mPCT). Panel (**A**) shows that the expression of ECFP/Ang II as a surrogate of intracellular Ang II stimulated NHE3 protein expression in wild-type mPCT cells, and this response was significantly attenuated by losartan, an AT_1_ receptor blocker, and PD123319, an AT_2_ receptor antagonist. Panel (**B**) shows that the expression of ECFP/Ang II had insignificant effects in *Agtr1a*^-/-^ mPCT cells without or with losartan or PD123319 treatment. ** *p* < 0.01 vs. control; ^++^
*p* < 0.01 vs. ECFP/ANG II.

**Figure 5 cells-12-01492-f005:**
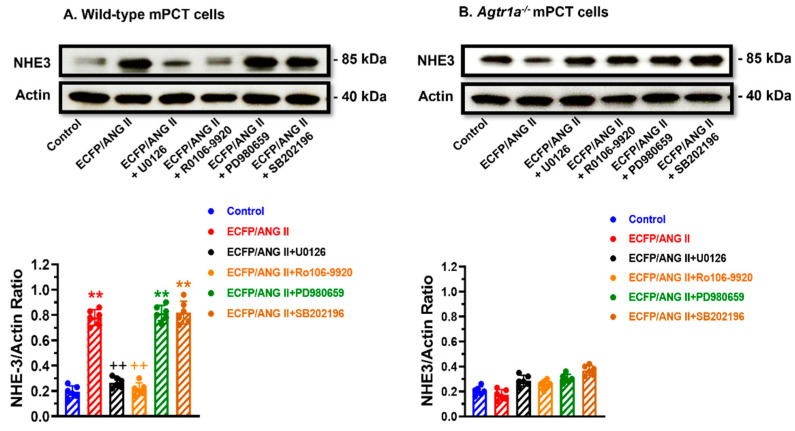
The roles of the MAP kinase and NF-κB signaling pathways in mediating ECFP/Ang II-induced NHE3 expression in wild-type and *Agtr1a*^-/-^ mPCT cells. Panel (**A**) shows that in wild-type mPCT cells, ECFP/Ang II stimulated NHE3 expression significantly, and the response was attenuated by the MEK1/MEK2 kinase inhibitor U0126 and the NF-κB activation inhibitor Ro 106–9920, respectively. However, the MEK inhibitor PD 980659 and the p38 MAP kinase inhibitor SB202196 failed to attenuate the effect of ECFP/Ang II on NHE3 expression. Panel (**B**) shows that in *Agtr1a*^-/-^ mPCT cells, ECFP/Ang II failed to stimulate NHE3 expression, and the inhibitors of the MAP kinases and NF-κB signaling pathways had no significant effects on NHE3 expression. ** *p* < 0.01 vs. control WT mPCT cells. ^++^ *p* < 0.01 vs. WT mPCT cells transfected with ECFP/Ang II.

**Figure 6 cells-12-01492-f006:**
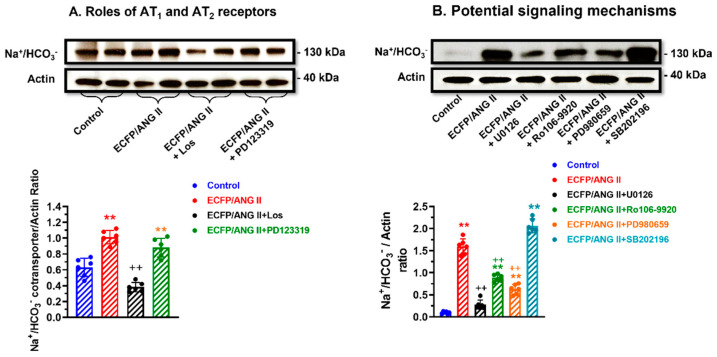
The roles of AT_1_ and AT_2_ receptors, the MAP kinases, and NF-κB signaling pathways in mediating ECFP/Ang II-induced Na^+^/HCO_3_^-^ cotransporter expression in wild-type mPCT cells. Panel (**A**) shows that ECFP/Ang II significantly increased Na^+^/HCO_3_^-^ expression, and the response was attenuated by losartan but not by PD123319, suggesting a dominant role of AT_1_ receptors in mPCT cells. Panel (**B**) shows that the MEK1/MEK2 kinase inhibitor U0126, the NF-κB activation inhibitor Ro 106–9920, and the MEK inhibitor PD 980659 attenuated the effects of ECFP/Ang II on expression, but the p38 MAP kinase inhibitor SB202196 had no effect on Na^+^/HCO_3_^-^ expression. ** *p* < 0.01 vs. control WT mPCT cells. ^++^ *p* < 0.01 vs. WT mPCT cells transfected with ECFP/Ang II.

**Figure 7 cells-12-01492-f007:**
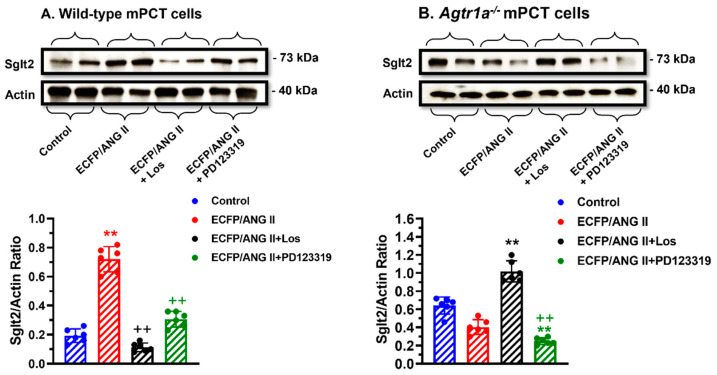
The roles of AT_1_ and AT_2_ receptors in mediating ECFP/Ang II-induced sodium and glucose cotransporter 2 (SGLT2) expression in wild-type and *Agtr1a*^-/-^ mPCT cells. Panel (**A**) shows that ECFP/Ang II significantly increased SGLT2 expression, and the response was attenuated by both losartan and PD123319, supporting an important role of AT_1_ and AT_2_ receptors in wild-type mPCT cells. Panel (**B**) shows that in the absence of AT_1a_ receptors, ECFP/Ang II had no effect on SGLT2 expression in *Agtr1a*^-/-^ mPCT cells. However, losartan increased SGLT2 expression, whereas PD123319 decreased SGLT2 expression in *Agtr1a*^-/-^ mPCT cells, suggesting an important role for AT_2_ receptors in SGLT2 expression in *Agtr1a*^-/-^ mPCT cells. ** *p* < 0.01 vs. control WT or *Agtr1a*^-/-^ mPCT cells. ^++^ *p* < 0.01 vs. WT or *Agtr1a*^-/-^ mPCT cells transfected with ECFP/Ang II and treated with losartan.

**Figure 8 cells-12-01492-f008:**
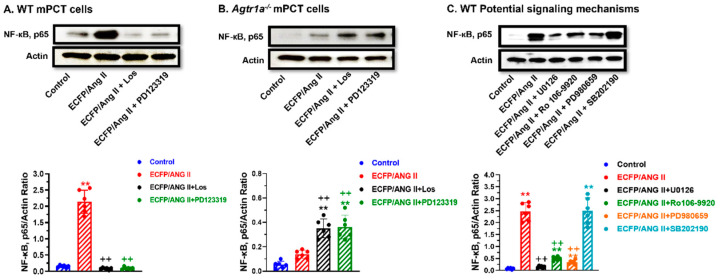
The roles of AT_1_ and AT_2_ receptors, the MAP kinases, and NF-κB signaling pathways in mediating ECFP/Ang II-induced NF-κB, p65 expression in wild-type and *Agtr1a*^-/-^ mPCT cells. Panel (**A**) shows that ECFP/Ang II increased NF-κB, p65 expression in wild-type mPCT cells, and the response was attenuated by both losartan and PD123319, supporting an important role of AT_1_ and AT_2_ receptors in mediating ECFP/Ang II-induced NF-κB, p65 expression in wild-type mPCT cells. Panel (**B**) shows that ECFP/Ang II alone had no significant effect on NF-κB, p65 expression in *Agtr1a*^-/-^ mPCT cells, but both losartan and PD123319 potentiated this response. Panel (**C**) shows that in wild-type mPCT cells, the effect of ECFP/Ang II on NF-κB, p65 expression was attenuated by the MEK1/MEK2 kinase inhibitor U0126, the NF-κB activation inhibitor Ro 106–9920, and the MEK inhibitor PD 980659, respectively. However, the p38 MAP kinase inhibitor SB202196 had no effect on ECFP/Ang II-induced NF-κB, p65 expression in wild-type mPCT cells. ** *p* < 0.01 vs. control WT or *Agtr1a*^-/-^ mPCT cells. ^++^ *p* < 0.01 vs. WT mPCT cells transfected with ECFP/Ang II, or *Agtr1a*^-/-^ mPCT cells transfected with ECFP/ANG II.

**Figure 9 cells-12-01492-f009:**
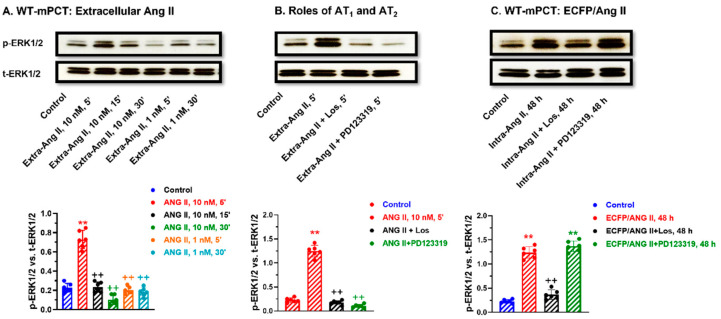
Comparisons of AT_1_ and AT_2_ receptor-mediated, extracellular Ang II- versus intracellular ECFP/Ang II-induced MAP kinases ERK1/2 activation in wild-type mPCT cells. (**A**) Time- and dose-dependent responses to extracellular Ang II stimulation with a peak response for 10 nM at 5 min. (**B**) The effect of losartan (an AT_1_ blocker) and PD123319 (an AT_2_ blocker). (**C**) The effect of losartan (an AT_1_ blocker) and PD123319 (an AT_2_ blocker) on intracellular ECFP/Ang II-induced MAP kinases ERK1/2 activation. These data suggest that both extracellular and intracellular Ang II activate MAP kinases ERK1/2 signaling in wild-type mPCT cells. ** *p* < 0.01 vs. control. ^++^ *p* < 0.01 vs. ECFP/Ang II.

**Figure 10 cells-12-01492-f010:**
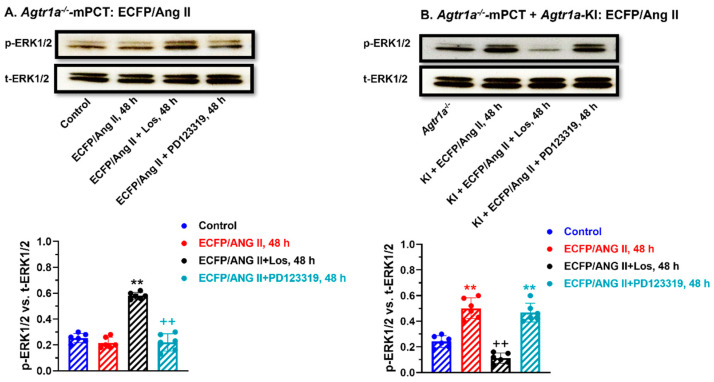
AT_1_ (AT_1a_) receptor-mediated, intracellular ECFP/Ang II-induced activation of the MAP kinases ERK1/2 signaling pathway in *Agtr1a*^-/-^ mPCT cells without or with the knock-in of a full-length wild-type mouse AT_1a_ receptor. ** *p* < 0.01 vs. control; ^++^ *p* < 0.01 vs. ECFP/Ang II + Los (**A**) or ECFP/Ang II (**B**), respectively.

**Figure 11 cells-12-01492-f011:**
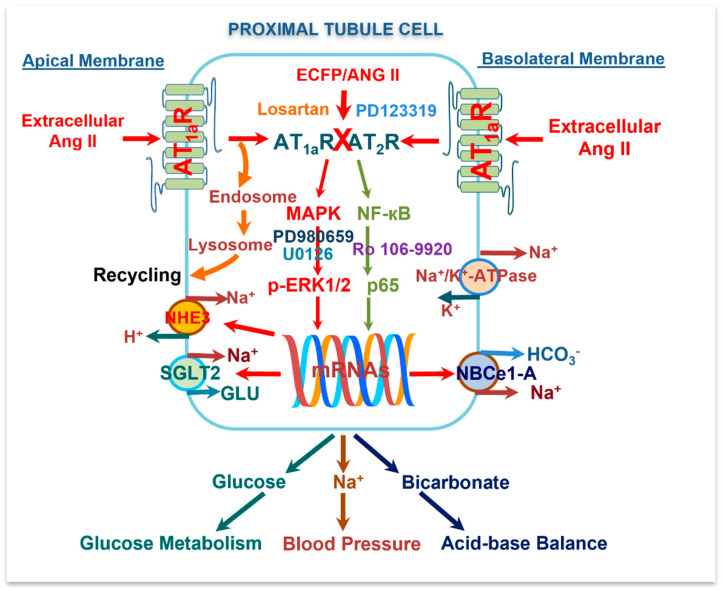
The schematic diagram summarizing the potential Ang II/AT_1_ (AT_1a_) and AT_2_/MAP kinases ERK1/2 and NF-κB signaling mechanisms by which extracellular and intracellular Ang II activates cell surface and intracellular AT_1_ (AT_1a_) and AT_2_ receptors to induce important transcriptional responses, i.e., the expression of NHE3 antiporter, SGLT2, and Na^+^/HCO_3_^-^ cotransporters in mouse proximal tubule cells. The results of the present study suggest that intracellular Ang II and its receptors in the proximal tubules of the kidney may play important physiological roles in normal blood pressure regulation, glucose metabolism, and acid-base balance.

## Data Availability

The authors agree to make all methods and materials including the protocols for genotyping, surviving and non-surviving mouse surgical protocols, experimental design in mice or in cell culture studies, and all supporting raw data available to other researchers upon requests.

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
