# Peer review of "Intracellular Angiotensin II Stimulation of Sodium Transporter Expression in Proximal Tubule Cells via AT1 (AT1a) Receptor-Mediated, MAP Kinases ERK1/2- and NF-кB-Dependent Signaling Pathways"

_cells, 2023, doi:10.3390/cells12111492_

Round 1

Reviewer 1 Report

The current study by Li and Zhuo investigates the tubular Angiotensin II (Ang II) system in the mouse kidney and a mouse proximal tubule cell line.  Their previous studies identified uptake of Ang II in the kidney (rat, mouse, human), predominantly in the proximal tubules and intracellular expression of Ang II.  To distinguish the extracellular effects of Ang II from receptor-mediated internalized Ang II, authors have also shown that a proximal tubule-directed Ang II fusion protein is expressed exclusively in the tubular epithelium and has similar effects to extracellular Ang II including an increase in blood pressure associated with greater expression of NHE3.   The present study utilizes the SGLT2 Ang II fusion vector in mouse tubule cells to establish the signaling pathways involved in NHE3, Na/HCO3, SGLT2 and NFkB expression or activation.   Overall, these are interesting and novel data on the intracellular actions of Ang II (iAng II) in the regulation of these key Na+ transporters and signaling pathways.  However, there are several concerns with study regarding the effects of the AT2R antagonist as compared to the AT1R KO cells that require additional approaches to discern a true AT2R action.  

1.      Much of the data up through Figures 1-3 has already been extensively described in their previous publications and reviews.  These figures should be deleted from the current study. 

2.      It is not clear to the reviewer that one can equate expression of iAng II with internalization of the Ang II-AT1R complex as the latter is destined to endosomes and lysosomes for Ang II degradation and recycling of the AT1R.   Intracellular expression of the fusion Ang II is unclear as to its intracellular trafficking and compartmentalization in the current study.  Can the authors further clarify these pathways? 

3.      The data in Figure 4 show that PD abolished the effect of intracellular Ang II (Ang II) on NHE3 expression, yet the AT1a knockout also abolished the iAng II effect (Fig 4A vs 4B).   But how can the authors equate this to a specific AT2R effect?   Either the PD compound is not selective or the AT1aR KO essentially depletes the cells of AT2R.  Authors must utilize AT2R KO cells or use siRNA approaches to more definitely discern these apparent AT2R effects in their cells rather than solely rely on the PD compound. 

4.      In Figure 5, authors screen various kinase inhibitors to show that MEK pathways blocks the iAng II-AT1R effect.  However, how does this effect occur?  What intracellular organelle or compartment is Ang II binding to elicit MEK activation? 

5.      Authors distinguish iAng II stimulation of MEK, but not ERK pathways in NHE3 stimulation.  What downstream targets is activation of MEK linked to for NHE3 regulation?    

6.      In Figure 6, the AT1R antagonist Losartan (LOS) reduced NaHCO3 markedly below baseline.   What effect does LOS have on WT cells?   Is this an extracellular or intracellular effect of LOS on the AT1R?  Do these cells generate endogenous Ang II that is secreted or resides intracellularly?  

7.      In Figure 7,  both LOS and PD block iAng II stimulation of SGLT2; as well as KO of AT1aR.   LOS stimulates SGLT2 in the AT1R KO cells with iAng II, but how does this occur in the KO cells with no AT1R?   Again, authors should use additional approaches to discern an AT2R effect, as well as determine whether AT2R density changes in WT and AT1R KO cells. 

8.      The data in Figure 7 suggest that the AT2R is anti-natriuretic as iAng II stimulates SGLT2 in the tubule cells, particularly with LOS treatment.   Doesn’t this directly contrast with the functional data of the AT2R in the kidney by Cary and colleagues, as well as other investigators suggesting that AT2R increase Na+ excretion to reduce blood pressure?  Again, why is there a LOS effect in AT1 KO cells?  Also, does the iAng II direclty bind AT2R or is this converted to Ang III to recoginze the AT2R?

9.      The data in Figures 8-9 show individual western blots that lack quantification of the data.   As shown, these data are impossible to judge the significant effects of treatment and a sufficient N with statistical analysis must be included.   

10.  Specifically, what studies have shown that  AT2R are linked to MAPK activation?   Previous data suggest that AT2R increase cellular phosphatase activity that would appear to oppose MAPK phosphorylation (doi: 10.1042/bj3250449;  doi: 10.1291/hypres.24.385; DOI: 10.1152/ajpheart.1998.275.3.H906)

11.  Text font changes throughout paper needs correction.

12.  Provide the number of cell experiments for each figure.

13.  A cartoon depicting the intracellular Ang II and associated signaling pathways would be helpful in the overall presentation of the data.    

Author Response

Dear Reviewer:

We would like to thank you for your careful and constructive reviews, comments, and expert recommendations on our manuscript. Just let you know that we have carefully considered your critiques and tried our best to revise and improve our manuscript accordingly. Please accept our sincere apology if we have not satisfactorily answered (or disagree with) your critiques, and implemented your recommendations, which may be simply due to the differences of our opinions. Below are our responses to your reviews:

Reviewer Comment #1: Much of the data up through Figures 1-3 has already been extensively described in their previous publications and reviews.  These figures should be deleted from the current study. 

Authors' Responses: We understand the Reviewer's point of views. However, we may have to disagree with the reviewer's recommendation. In our humble views, these data are an important and key part of this study that provides scientific rationales to test our hypothesis on whether extracellular angiotensin is taken up by proximal tubule cells to act as an intracellular peptide to stimulate the expression of sodium transporter or cotransporters. Without these in vivo data showing that extracellular angiotensin II is taken up by proximal tubule cells, there will be the lack of the scientific rationale or justifications to express an intracellular angiotensin II fusion protein to study the effects of intracellular angiotensin II on the sodium transporter expression in this manuscript? For example, we have already known that angiotensin II acts on AT1 receptor to increase blood pressure, induce salt retention, and cause cardiac hypertrophy for decades, yet we are still seeing these data are repeatedly shown in recently published articles in most if not all scientific journals. 

Reviewer Comment #2: It is not clear to the reviewer that one can equate expression of iAng II with internalization of the Ang II-AT1R complex as the latter is destined to endosomes and lysosomes for Ang II degradation and recycling of the AT1R.   Intracellular expression of the fusion Ang II is unclear as to its intracellular trafficking and compartmentalization in the current study.  Can the authors further clarify these pathways? 

Authors' Responses: We agree that the reviewer has a "valid" point that one can not equate expression of iAng II with internalization of the Ang II-AT1R complex. We do not want to equate these two being the same as no one in reality can truly separate intracellularly generated Ang II from extracellular Ang II internalized with cell surface AT1  receptors. The reviewer argues that extracellular Ang II-AT1R complex is destined to the endosome/lysosome degradation pathways, which is indeed the current dogma of the classic GPCR pharmacology. However, in our humble views this dogma is not completely accurate as we and many others have repeatedly demonstrated in cell culture and in vivo studies showing extracellularly or systemically administered or infused fluorescent or radiolabeled Ang II was not only internalized into the endosome/lysosome pathways, but also trafficked to other intracellular organelles such as endoplasmic and reticulum, mitochondria, and the nuclei. When people are saying that extracellular Ang II-AT1R is destined to the endosome/lysosome/degradation pathways, this is because previous experiments were done for 5 min or 30 min to track the the endocytic pathways. However, if the same experiments last for hours or fluorescein or radiolabled Ang II is infused for hours or weeks, one will readily find internalized Ang II in the mitochondria, endoplasmic reticulum, and nuclei. Arterial blood pressure will continue to increase as long as Ang II infusion continues; and one will find no Ang II-AT1R desensitization if Ang II-AT1R is indeed destined only to go to the endosome/lysosome pathways for degradation. The point we would like to make here is not to argue with the reviewer, instead we based on these previous findings to test our hypothesis in this manuscript. 

The Reviewer Comment #3: The data in Figure 4 show that PD abolished the effect of intracellular Ang II (Ang II) on NHE3 expression, yet the AT1a knockout also abolished the iAng II effect (Fig 4A vs 4B).   But how can the authors equate this to a specific AT2R effect?   Either the PD compound is not selective or the AT1aR KO essentially depletes the cells of AT2R.  Authors must utilize AT2R KO cells or use siRNA approaches to more definitely discern these apparent AT2R effects in their cells rather than solely rely on the PD compound. 

Authors' Responses: We thanks the reviewer for this expert comment. Indeed, although PD123319 is widely used as a specific antagonist for AT2 receptors, it also blocked the effect of iAng II in cell culture studies, in some other studies, PD123319 also blocked Ang II-induced cellular responses. This is surprising to us as well as AT2 receptors are supposed to counteract the effects of AT1 (AT1a) receptor activation. There is a potential possibility that at high pharmacological concentrations, PD123319 may act as as an AT1 receptor antagonist. The reviewer recommended to use AT2R-KO mouse proximal tubule cells or AT2 receptor-siRNAs, which is a great idea. Unfortunately, no AT2R-dificent mouse proximal tubule cells are available to do these experiments. Second, the use of AT2R-siRNA to silent AT2R in mouse proximal tubule cells may also pose a problem, as AT2R accounts for only less than 10% of total Ang II receptors in proximal tubule cells, and its silence may not easily reveal any effects in the presence of predominant, i.e., ~90%, AT1 (AT1a) receptors in these cells. To address the reviewer's comments, we have tone down the statement and revised the manuscript accordingly.    

Reviewer's Comment #4: In Figure 5, authors screen various kinase inhibitors to show that MEK pathways blocks the iAng II-AT1R effect.  However, how does this effect occur?  What intracellular organelle or compartment is Ang II binding to elicit MEK activation? 

Responses: The reviewer made a very expert comment on how the inhibitors of various MEK pathways block the iAng II-AT1R effect, and which organelle or compartment iAng II binds and activates these MEK kinases. As we used the whole-cell approach, we can not accurately identify the intracellular organelles or compartments in which iAng II activates these MEK kinases. However, we do know that AT1 (AT1a) receptors and iAng II are present in most if not all intracellular organelles, where iAng II may bind AT1 (AT1a) receptors to activate MEK kinases intracellularly, which may be blocked by the MEK inhibitors.  

Reviewer' Comment #5: Authors distinguish iAng II stimulation of MEK, but not ERK pathways in NHE3 stimulation.  What downstream targets is activation of MEK linked to for NHE3 regulation? 

Authors' responses: ERK1/2 is supposed to be the downstream target of the activation of the MAKP/ERK pathway by iAng II, and p-ERK1/2 is the link to NHE3 regulation.    

Reviewer Comment #6: In Figure 6, the AT1R antagonist Losartan (LOS) reduced NaHCO3 markedly below baseline.   What effect does LOS have on WT cells?   Is this an extracellular or intracellular effect of LOS on the AT1R?  Do these cells generate endogenous Ang II that is secreted or resides intracellularly?  

Authors' Responses: Thanks for this interest comment. Losartan is well recognized and widely used as a very specific antagonist for AT1R. Losartan is not only bind to the cell surface AT1 receptors but we have also previously shown that losartan is also internalized into the proximal tubule cells. Proximal tubules cells also express nearly all major components of the renin-angiotensin system especially angiotensinogen, renin, angiotensin-converting enzyme, and AT1R (AT1aR). Thus it is expected that Ang II is generated both intracellularly and extracellularly. We believe that losartan blocks both intracellular and extracellular Ang II-induced effects on NaHCO3 expression.  

Reviewer Comment #7: In Figure 7,  both LOS and PD block iAng II stimulation of SGLT2; as well as KO of AT1aR. LOS stimulates SGLT2 in the AT1R KO cells with iAng II, but how does this occur in the KO cells with no AT1R?   Again, authors should use additional approaches to discern an AT2R effect, as well as determine whether AT2R density changes in WT and AT1R KO cells. 

Authors' Responses: We have explained above the difficulty of performing the additional experiments recommended by the reviewer to discern the roles of AT2R on other Na+ transporter or cotransporters. Still, we accept the reviewer's recommendation to use AT2R-KO cells or AT2R-siRNA approach to discern the effect of AT2R in future studies, but not for this manuscript due to the time-sensitive constrains.  

Reviewer Comment #8: The data in Figure 7 suggest that the AT2R is anti-natriuretic as iAng II stimulates SGLT2 in the tubule cells, particularly with LOS treatment.  Doesn’t this directly contrast with the functional data of the AT2R in the kidney by Cary and colleagues, as well as other investigators suggesting that AT2R increase Na+ excretion to reduce blood pressure?  Again, why is there a LOS effect in AT1 KO cells?  Also, does the iAng II direclty bind AT2R or is this converted to Ang III to recoginze the AT2R?

Authors' Responses: The reviewer has a very interesting question regarding the roles of AT2R in the proximal tubules. It is correct that Carey's group has consistently shown that activation of AT2R by Ang II or its metabolite Ang III induced a natriuretic response and decreased blood pressure in the rats and mice. However, we are not aware that other labs have performed similar studies and reported similar natriuretic responses. Indeed, global AT2R knockout in mice showed very little blood pressure effects or antinatriuretic responses, nor did AT2R blockade with PD123319 in rats and mice. Most if not all studies showed the effects of AT2R only in the presence of AT1R blockade by losartan or candesartan. In the present study, the expression of iAng II in the proximal tubule cells had no effect on Sglt2 expression in AT1R-KO cells, whereas losartan treatment augmented Sglt2 expression and PD123319 blocked this effect. These results are not consistent with the studies of Carey's group, but are consistent with the notion that in the absence of AT1a receptors (AT1a-KO) or AT1b receptors, iAng II activates AT2R to induce Sglt2 expression in mouse proximal tubule cells. Please see doi: 10.1097/HJH.0b013e328349ae0d for AT2R mediates vasocontriction in human blood vessels.

Reviewer Comment #9: The data in Figures 8-9 show individual western blots that lack quantification of the data.   As shown, these data are impossible to judge the significant effects of treatment and a sufficient N with statistical analysis must be included.   

Authors' Responses: We apologize for the oversight and thank the reviewer's constructive comment. We have included semiquantitative data from at least 6 Western blot samples in the revised manuscript.  

Reviewer Comment #10: Specifically, what studies have shown that  AT2R are linked to MAPK activation?  Previous data suggest that AT2R increase cellular phosphatase activity that would appear to oppose MAPK phosphorylation (doi: 10.1042/bj3250449;  doi: 10.1291/hypres.24.385; DOI: 10.1152/ajpheart.1998.275.3.H906)

Authors' Responses: Thank you for the reviewer's comments. As a matter of fact, both inhibition and activation of the MAP kinases ERK1/2 by Ang II via AT2R have been reported in the literature using PD123319 in many different cells or tissues. Normally, investigators would like to report and accept the dogma that activation of AT2R is to oppose the effect of AT1R activation. In reality, many studies have shown that AT2R blockade or activation by Ang II either inhibited, increased, or had no effect on MAP kinases ERK1/2 signaling. We list here just a few studies to show that Ang II may bind and activate AT2R to inhibit, increase or have no effect on MAP Kinases ERK1/2 in different cells or tissues. https://doi.org/10.1016/j.bcp.2006.03.018; DOI: 10.1074/jbc.271.26.15635; https://doi.org/10.1016/j.regpep.2004.10.005; https://doi.org/10.1016/j.regpep.2004.10.005; https://doi.org/10.1152/ajprenal.00455.2011 (glomerular mesangial cells); DOI: 10.1016/j.npep.2011.07.002 (Astrocytes); and many more if there are spaces.

Reviewer Comment #11: Text font changes throughout paper needs correction. 

Authors' Response: Thank you very much, and we have made text font consistent throughout the revised manuscript accordingly.

Reviewer Comment #12: Provide the number of cell experiments for each figure.

Authors' Response: We have stated clearly that at least 6 cell sample experiments were performed and analyzed and presented in the revised manuscript as suggested by the reviewer.

Reviewer Comment #13: A cartoon depicting the intracellular Ang II and associated signaling pathways would be helpful in the overall presentation of the data. 

Author's Response: As suggested by the reviewer, we have included a schematic diagram depicting the iAng II-induced signaling pathways in the revised manuscript as Figure 11.

Thank you very much for your expert and constructive comments and recommendations to help us improve the manuscript accordingly.

Reviewer 2 Report

This is an interesting paper demonstrating novel intracellular roles of Ang II via intracellular (?) AT1 receptor activation in renal proximal tubule cells in vivo and in vitro.  The reviewer has several minor concerns and suggestions.

Please include scattered plots in all bar graphs.

Please include more upper and lower area of the bans in Western blotting.

Figure 4A data also suggest important role of the AT2 receptor.  Please explain.

Possibility should be included that PD980659 may show inhibitory effect if higher concentration was utilized.

Single blot presentation in Fig 8 and 9 is a concern.  These data may be deleted.

Please discuss the possibility of intracellular Ang II activates extracellular AT1 or AT2 receptor.

Please discuss whether extracellular Ang II equally or distinctly regulates the transporters.

Author Response

Dear Reviewer: We would like to thank you for your kind reviews, expert comments, and constructive suggestions on our manuscript.  We have carefully reviewed and considered your comments and suggestions in revising our manuscript as follows.

Reviewer Comment #1: Please include scattered plots in all bar graphs.

Author's Response: Thank you for the suggestion and we have revised all bar graphs with scattered plots in the revised manuscript.

Reviewer Comment #2: Please include more upper and lower area of the bans in Western blotting.

Authors' Response: Although this is a great suggestion, however, re-cutting all Western blots from all original Western blot images to redo all figures involves a lot of work without improving the quality of the Western blot images or data. Furthermore, as requested we have submitted to the journal's Home Website all original Western blot films/images that were used for all figures with Western blot studies.

Reviewer Comment #3: Figure 4A data also suggest important role of the AT2 receptor.  Please explain.

Author's Responses: Indeed, since the expression of intracellular Ang II fusion protein increased NHE3 expression, whereas not only losartan (the AT1 receptor blocker) but also PD123319 (the AT2 receptor blocker) blocked this stimulatory effect, these data suggest that AT2 receptors may play an important role in the regulation of NHE3 expression in mouse proximal tubule cells. However, these data are different from previous studies that activation of the AT2 receptor by extracellular Ang II counteracts the effect of the AT1 receptor activation by extracellular Ang II, which have not been reported previously by others.

Reviewer Comment #4: Possibility should be included that PD980659 may show inhibitory effect if higher concentration was utilized.

Authors' Response: We have included your comment in the discussion on high concentrations of PD980659 in the revised manuscript.

Reviewer Comment #4: Single blot presentation in Fig 8 and 9 is a concern.  These data may be deleted.

Authors' Responses: We have performed more experiments and revised these figures as new scatter plots with the number of the cell experiments in the revised manuscript.

Reviewer Comment #5: Please discuss the possibility of intracellular Ang II activates extracellular AT1 or AT2 receptor.

Authors' Responses: This is a great suggestion from the reviewer. We have discussed the possibilities that intracellular Ang II may activate extracellular AT1 or AT2 receptors in the revised manuscript.

Reviewer Comment #6: Please discuss whether extracellular Ang II equally or distinctly regulates the transporters.

Authors' Responses: Thank you for the suggestions. We have discussed that extracellular Ang II equally or distinctly regulates the expression of Na+ transporters or cotransporters in the revised manuscript as suggested by the reviewer.

Reviewer 3 Report

"Intracellular angiotensin II stimulation of sodium transporter expression in proximal tubule cells via AT1 (AT1a) receptor-mediated, MAP Kinases ERK1/2- and NF-кB-dependent signaling pathways" by: Xiao C. Li , and Jia Zhuo

Please explain why you use radioactive isotope and fluorophore compound.

Please explain why you used rats, instead of wild type mice for the AT1 (AT1a) receptor-mediated uptake of [125I]-labeled Ang II.

Please explain how you differentiate endogenous angiotensin II from exogenous, regarding the effect? Angiotensin II production inhibition would help addressing the issue, why it was not inhibited? Please include the subject in discussion.

Endogenous production of angiotensin II was not evaluated. Information regarding angiotensin II production in renal cells is lacking, please include information regarding these subjects and discuss.

Please review English spell, grammatics, and letter size Please italize in vivo and in vitro.

Please provide source information regarding used compounds and equipment.

Figure 2. Bars on figure 2 are differently arranged than figure 1, please homogenize. 

Figure 2 legend mentions: ... WT and Agtr1a-/- mice, expressed as relative fluorescence intensity with WT mice at 100. What is the meaning of the number "100"?  

Figures 6 and 7. Western blots lack information regarding the content of lines

Figure 8. Why p65 expression increases in Agtr1a-/- mice treated with losartan? It is required to show a graph expressing the results.

Figure 9. Please, clearly explain on figure legend the labels of the lines.

Author Response

Dear Reviewer:

We would like to thank you for your kind reviews, expert comments, and constructive suggestions on our manuscript.  We have carefully reviewed and considered your comments and suggestions in revising our manuscript as follows.

Reviewer Comment #1: Please explain why you use radioactive isotope and fluorophore compound.

Author's Responses: Thanks for the comments. The purpose of using radioactive isotope- or fluorescein-labeled Ang II in the present study was to determine and clearly visualize whether extracellular Ang II is taken up by the proximal tubules of the kidney (i.e., seeing is believing). This is especially and very relevant to test our hypothesis and provides a strong scientific rationale for the expression of an intracellular Ang II fusion protein in cultured mouse proximal tubule cells to determine whether intracellular Ang II regulates the expression of Na+ transporters or cotransporters in the kidney.  

Reviewer Comment #2: Please explain why you used rats, instead of wild type mice for the AT1 (AT1a) receptor-mediated uptake of [125I]-labeled Ang II.

Authors' Responses: The rationale for using both rats and mice for the uptake experiments is for a better visualization of the uptake response in the rat proximal tubules (i.e., one rat kidney is about 10-fold larger than a mouse kidney), and for determine the role of AT1a receptors using both wild-type and Agtr1a-/- mice. These studies are complementary and supportive of using each species. Furthermore, there is currently no good rat Agtr1a-/- model for the proposed studies. 

Reviewer Comment #3: Please explain how you differentiate endogenous angiotensin II from exogenous, regarding the effect? 

Author's Responses: In the present study, we did not apply extracellular Ang II in the medium to bind and stimulate cell membrane Ang II receptors. Instead, we expressed an intracellular Ang II fusion protein which has been shown to not secreted into the medium or extracellularly, the blood, or the urine (DOI: 10.1152/ajprenal.00329.2010). This suggests that the expressed intracellular Ang II is confined intracellularly and only stimulates intracellular AT1 or AT2 receptors. 

Reviewer Comment #4:  Angiotensin II production inhibition would help addressing the issue, why it was not inhibited? Information regarding angiotensin II production in renal cells is lacking, please include information regarding these subjects and discuss.

Authors' Responses: Thanks for the suggestion, however, we have previously determined and reported Ang II production in cultured proximal tubule cells. These additional data and information would be redundant in the revised manuscript.

Reviewer Comment #5: Please review English spell, grammatics, and letter size Please italize in vivo and in vitro.

Authors' Responses: As the reviewer suggested, we have rechecked, reviewed, and corrected all misspells, grammatic errors, and letter size throughout the manuscript. In vitro and In Vivo have been italized as suggested. 

Reviewer Comment #6: Please provide source information regarding used compounds and equipment.

Authors' Responses: The requested information for compounds or equipment used in the present study has been included in the revised manuscript.

Reviewer Comment #7: Figure 2. Bars on figure 2 are differently arranged than figure 1, please homogenize. 

Authors' Responses: Thank you for the comment and these two bars have been homogenized and kept consistent.

Reviewer Comment #8: Figure 2 legend mentions: WT and Agtr1a-/- mice, expressed as relative fluorescence intensity with WT mice at 100. What is the meaning of the number "100"?  

Authors' Response: Normally, it is very difficult to accurately quantitate the fluorescence level in cells or tissues. We set the relative or arbitrary fluorescence intensity (RFU) in the kidney of wild-type mice at 100% for a direct comparison with that of AT1a-KO mice. 

Reviewer Comment #9: Figures 6 and 7. Western blots lack information regarding the content of lines

Authors' Responses: These oversights have been corrected as you suggested.

Reviewer Comment #10: Figure 8. Why p65 expression increases in Agtr1a-/- mice treated with losartan? It is required to show a graph expressing the results.

Authors' Responses: Thank you for your expert comment. We are not completely not sure why p65 expression increases in Agtr1a-/- mouse proximal tubule cells. This suggests that in the absence of AT1a and AT1b, AT2 receptors may be activated by intracellular Ang II, which has been shown by extracellular Ang II previously by others. However, this effect in Agtr1a-/- cell was not blocked by PD123319, suggesting more mechanistic studies may be necessary in later studies. 

Reviewer Comment #11: Figure 9. Please, clearly explain on figure legend the labels of the lines.

Authors' Responses: Yes, we have added labels of the lines for the Western blot images, and bar graphs with scatter plots in the revised Figure 9 to more clearly explain the results in the revised manuscript.

Again, thank you very much for helping us revise and further improve the manuscript.